# Fossilized solidifications fronts in the Bushveld Complex argues for liquid-dominated magmatic systems

Willem Kruger[1] & Rais Latypov [1✉]

Chemical differentiation of magma on Earth occurs through physical separation of liquids and crystals. The mechanisms of this separation still remain elusive due to the lack of information on solidification fronts in plutonic magmatic systems. Here, we present records of fossilized solidification fronts from massive magnetitites of the Bushveld Complex in South Africa, obtained by two-dimensional geochemical mapping on field outcrops. The chemical zoning patterns of solidification fronts indicate that nucleation and crystallization occur directly at the chamber floor and result in near-perfect fractionation due to convective removal of a compositional boundary layer from in situ growing crystals. Our data precludes the existence of thick crystal mushes during the formation of massive magnetitites, thus providing no support for the recent paradigm that envisages only crystal-rich and liquid-poor mushy reservoirs in the Earth's crust.

[1] School of Geosciences, University of the Witwatersrand, Johannesburg, South Africa. ✉email: Rais.Latypov@wits.ac.za

At the heart of magma differentiation on Earth and other terrestrial planets, irrespective of whether this occurs in shallow magma chambers[1–8], magma oceans/impact melt sheets[9,10] or planetary cores[11,12] is a physical separation of chemically distinct crystals and liquids. Exactly how this separation takes place to cause chemical differentiation of magmas is a central question of modern volcanology and igneous petrology (Fig. 1). Three major physical mechanisms are commonly considered for this role in magmatic systems: gravity settling/floating of crystals on the chamber floor/roof that leaves behind an evolved liquid[2,3,5,8] (Fig. 1c), compaction of a mushy cumulate pile that squeezes the evolved liquid into the overlying magma reservoir[13–16] (Fig. 1d), and compositional convection in a mushy cumulate pile that drives out the buoyant evolved liquid into the overlying magma body[17–19] (Fig. 1e). These processes are well-established theoretically and experimentally, but what role they play in driving chemical differentiation of magmas in nature is still far from being fully understood.

Critical information necessary to address this issue is locked in the internal structure of solidification fronts; the partially crystalline zones of magma that occur along the margins of magmatic systems[6,20]. Drilling into historic Hawaiian lava lakes back in 60s–80s of the last century allowed the first documentation of the textural and chemical features of the solidification fronts in volcanic environments[20,21]. However, the solidification fronts in deep-seated, slow-cooling magma chambers is still poorly explored, and their physical and chemical dynamics are largely unknown because of their inaccessibility for direct petrological observations. Inferring chemical and physical properties of

solidification fronts from fossilized magma chambers is extremely difficult, with only a few prominent attempts being undertaken to overcome this challenge[1,22–24]. A major problem is that distribution coefficients ($D$) for compatible trace elements, such as Ni, Cr, and Sr, are not particularly high — they normally range from 2 to 15 for most rock-forming minerals[25]. As a result, these minerals are not chemically sensitive enough to record the evolution and propagation of solidification fronts in magmatic systems. In addition, the minerals are susceptible to re-equilibration with trapped liquid[26–28] which tends to obliterate the primary chemical patterns in solidification fronts of plutonic complexes.

Fortunately, there is one notable exception from this rule: the partitioning of Cr into magnetite, a rock-forming mineral that is common in the uppermost parts of mafic layered intrusions. A compilation of $D$ values for Cr in magnetite in silicate liquids shows that they may range from 20 to 300 (ref. [29]). A few experimental studies indicate that even higher values, from 69 to 620 (refs. [30–32]), are possible in basaltic systems. In more evolved melts, $D$ values may rise up to 1290 (ref. [33]). In addition, geochemical modelling shows that $D$ values need to be, at least, as high as 275 to 400 to reproduce the vertical variations in Cr across massive magnetitites of the Bushveld Complex[1,34,35]. The crystallization of magnetite is thus expected to cause a rapid depletion of basaltic melt in Cr which must be recorded in decreasing Cr abundance in massive magnetitites at the scale of solidification fronts. This principle has been first used by Cawthorn and McCarthy[1,24] and resulted in a discovery of steep gradients in Cr concentrations in vertical profiles across magnetitite layers and even individual magnetite crystals[36] from the

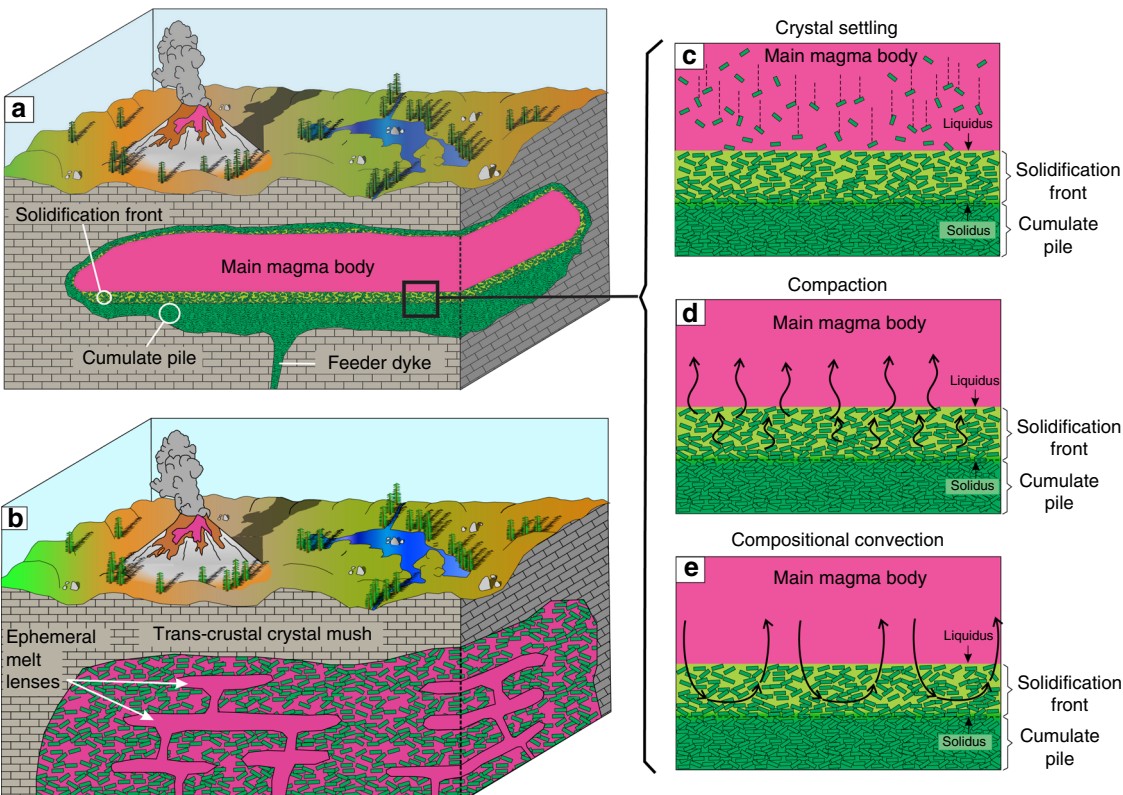

**Fig. 1 Potential magma reservoirs geometries and mechanisms for their evolution. a** The classical view of a melt-dominated, long-lived magmatic chamber situated underneath a volcano[1–8], compared with (**b**), a new concept that considers trans-crustal magmatic systems as composed of crystal-rich mushes with only small, short-lived melt lenses[16]. Differentiation of the magma can occur in several different ways: **c** crystal settling[2,3,5,8] whereby crystals grow in the interior of the magma chamber and settle towards the floor; **d** compaction of the cumulate pile[13–16], and **e** compositional convection within the cumulate pile[17–19] which forces a light evolved liquid out of mushy solidification fronts into the main magma body.

Bushveld Complex in South Africa. This finding led to an important realization that these layers are likely formed by in situ crystallization directly at the chamber floor[1,24,34,36]. Here, we build upon this pioneering work by undertaking a geochemical study of magnetitite layers in two dimensions using a handheld X-ray fluorescence spectrometer (pXRF). The study was motivated by an idea that the two-dimensional contours of the Cr decrease can be interpreted as Cr-isopleths that depict successive steps in the development of inward-growing solidification fronts in magma chambers. If so, the chemical patterns in magnetitite solidification fronts may provide a clue to the physical mechanisms that cause chemical differentiation of magmas in plutonic systems. Our two-dimensional geochemical study has confirmed a remarkable prediction by Cawthorn[37] that layers of magnetitite start forming via in situ growth nodes emerging at the contact with the footwall rocks. We show that during the growth of these nodes all rejected liquid components are effectively removed by compositional convection from the in situ growing crystals and transported into the main magma body. This results in effective adcumulus growth and inhibits the formation of crystal-rich mushes. We hypothesize that such compositional convection is induced by the instability of a chemical boundary layer around in situ growing crystals at the chamber floor and is one of the most effective mechanisms for melt fractionation in magmatic systems.

## Results

**Geochemical maps.** We have explored the local-scale variations in Cr distribution in massive magnetitite layers of the Bushveld Complex, the largest basaltic magma chamber in the Earth's crust[5,38], by two-dimensional geochemical mapping using pXRF (Supplementary Fig. 1; Supplementary Data 1) directly on field outcrops. Here, we present geochemical maps from the basal part of the most prominent magnetitite layer referred to as the Main Magnetite Layer (MML) (Fig. 2a). The near-monomineralic nature of this layer (Fig. 2b, c) implies that its chemical composition has been practically unaffected by a reaction with trapped liquid[26,36]. This implies that primary chemical patterns are likely preserved in solidification fronts of the MML. The layer was analysed in four separate areas, two of which show a planar basal contact (Fig. 3), one that contains a sub-rounded anorthosite autolith (Fig. 4), and another with a subvertical basal contact (Fig. 5). The irregularities in the floor (Figs. 4, 5) are thought to result from thermo-chemical erosion of the anorthositic footwall during magma replenishment of the chamber by basal flows as has been documented for other layers of the Bushveld Complex[39,40]. A key result of our study is the documentation of fine-scale domical structures with steep chemical gradients in Cr-content. These appear to be randomly distributed along the base of the MML and are characterized by a planar bottom with the highest Cr-content (up to 1.8–2.1 wt.%) that passes outwards in a concentric manner towards much lower Cr-contents (down to 1.0 wt.%) across a distance of only a few cm (Fig. 3b and d). Importantly, the structures with high Cr-content even occur below the anorthosite autolith (Fig. 4b). Moving upwards from the nodes, Cr-contours in magnetite turn into continuous sublayers with roughly planar contours. These sublayers tend to drape over the autolith (Fig. 4b) and develop roughly parallel to subvertical portions of a basal contact (Fig. 5b).

The first question to address is whether the high-Cr structures (Figs. 3–5) are primary magmatic features, or products of some secondary, non-magmatic processes superimposed on early formed, internally homogeneous magnetitites. One of the most common secondary processes is the action of late-stage hydrothermal fluids and/or melts that migrate upwards from the

underlying cumulates and can potentially affect the distribution of Cr within massive magnetitite[41]. However, such fluids or melts are expected to be highly depleted in Cr because of its partitioning into pyroxenes and magnetite in the underlying cumulate rocks. Therefore, the upward percolation of such fluids/melts and their reaction with magnetite may only cause its depletion in Cr, rather than the enrichment that is required to produce the high-Cr structures. Cr gradient in a single magnetite crystal also excludes the possible influence from late-stage interstitial fluids[36]. An alternative idea, namely the differential leaching of Cr from a magnetite layer by migrating fluids/melts, is also problematic because it would require a very peculiar flow pattern to form the domical high-Cr structures. It is also worth pointing out that a detailed geochemical study performed at our locality[42] failed to detect a notable change in Cr in magnetitite due to the activity of fluids coming from the underlying cumulate pile.

It seems, therefore, much more likely that the discovered high-Cr structures (Figs. 3–5) are of primary magmatic origin. The simplest explanation appears to be that they represent areas on the chamber floor where magnetite first starts nucleating and crystallizing outwards in a concentric manner. In this interpretation, we thus follow Cawthorn[37] who predicted the existence of such high-Cr structures in Bushveld's massive magnetitites and referred to them as in situ growth nodes. With further cooling, in situ growth nodes continue to spread laterally to cover the entire floor and eventually coalesce to form a layer with a planar surface (Supplementary Movie 1). Our results suggest that a planar solidification front may form after just 10–20 cm of crystallization, which explains why all magnetitite layers, including the thinnest ones (~10 cm), show planar top contacts. This tendency for the solidification front to become planar is also easily observed above the autolith (Fig. 4b). If our interpretation is correct, then these chemical structures (Figs. 3–5) are arguably the first ever records of the morphology of fossilized solidification fronts in deep-seated plutonic systems on Earth. The successive stages in in situ generation and subsequent upward propagation of these fronts are clearly indicated by Cr-isopleths.

The internal structure of the solidification fronts in the MML (Figs. 3–5) provides a unique opportunity to rigorously test the competing concepts of magma differentiation (Fig. 1). In situ growth nodes are clearly not compatible with gravity settling (Fig. 1c) as the latter implies a massive deposition of crystals from the overlying melt on the chamber floor to form a continuous layer, rather than a few small and highly localized Cr-rich nodes. The gravity concept is also at odds with the occurrence of the Cr-rich growth nodes below an autolith (Fig. 4) as well as the development of layers along the subvertical sidewalls (Fig. 5). These places are not appropriate for gravity-controlled accumulation of settling crystals. Similarly, in situ growth nodes are not consistent with compaction (Fig. 1d). The steep Cr-concentration gradients (>200 ppm/mm) in these nodes imply that chemical differentiation of melt became highly efficient almost immediately after the onset of magnetite crystallization. This dictates that rejected solute from every mm-thick 'shell' on in situ growing nodes must have been almost instantaneously transferred into the overlying melt to ensure its chemical evolution. Because compaction requires, at least, a few dozens to hundreds of metres of mush to operate[13,17,41], it cannot work in a magnetitite growth node less than a couple of centimetres thick. One may, perhaps, argue that compaction may squeeze interstitial liquid from a crystal mush below the magnetitite layer. However, recent work suggests that crystal mushes in layered intrusions are <4 m thick[43], and igneous cumulates are notably void of textural features indicating compaction[44]. In addition, deposition of a couple of centimetres of magnetite cannot provide sufficient loading to compact the underlying mush, thus precluding the

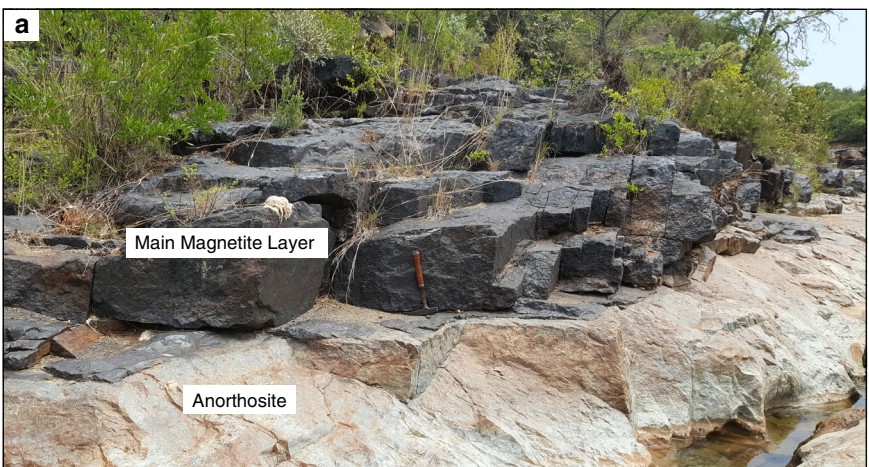

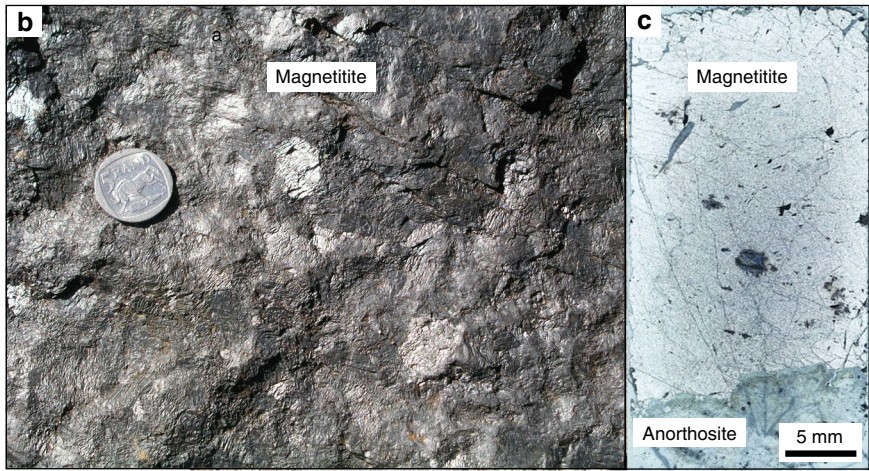

**Fig. 2 The Main Magnetite Layer at Magnet Heights from the Eastern Bushveld Complex. a** Field photograph of an outcrop of the MML with footwall anorthosites. The length of a hammer shaft is about 40 cm. **b** Field photograph of massive magnetitite from the MML. Sunlight reflected of individual titano-magnetite grains reveal the large grain size (typically more than 1 cm in size) and generally planar to slightly curved grain boundaries. Grains are typically polygonal with 120° triple-point grain boundary junctions, typical of monomineralic rocks in which textural equilibrium has been obtained. The coin has a diameter of 26 mm. **c** Photomicrograph in reflected light of the base of the MML including a portion of its underlying anorthositic footwall. Notice the near-perfect adcumulate and massive nature of the magnetite with virtually no intercumulus phases present.

transfer of the evolved liquid from below. The above objections can be equally well applied to compositional convection within a mush (Fig. 1e) because it has been estimated to require, at least, a 90 m thick mushy layer on the chamber floor for the onset of its operation[45]. The conventional models (Fig. 1c–e) thus fail the testing against the chemical features of magnetitite solidification fronts.

**Geochemical modelling.** Further constraints on a mechanism of magma differentiation during the formation of massive magnetite come from geochemical modelling. To explain the rapid upward depletion in Cr, previous studies[1,35] have indicated that the magnetitite could not have crystallized from the entire, ~1.5 km-thick column of resident melt in the chamber[1]. Instead, it was concluded that the magnetitites must have been crystallizing from a relatively thin isolated layer of melt at the chamber floor[1,35]. At issue is to understand which process is responsible for the formation of this basal layer of melt. In a closed system, it can be potentially produced by double-diffusive convection[35,46], liquid immiscibility[47–49], in situ fractionation of abundant plagioclase[50] or melt stagnation at the chamber floor[1,51]. In an open system, it can be most easily formed by a new pulse of a melt replenishing the chamber as a basal flow[52,53]. The dilemma can be addressed

by referring to the recent work on the Upper Zone of the Bushveld Complex that revealed a few prominent reversals in terms of Cr content in magnetite[54] and pyroxene[55] as well as in An-content in plagioclase and magnesium number of mafic silicates in the proximity of magnetite layers[54,55]. It has also been earlier reported that the basal part of the MML contains up to 240 times more Cr than pure magnetite separates from the underlying rocks[1]. These reversals in composition of cumulus minerals are clearly not consistent with the above-mentioned closed system models. The reversals are more compatible with the interpretation of the Upper Zone as an open system, with the magma additions into the chamber contributing to the formation of massive magnetitites[54–56]. Following these studies[54–56], we argue that the replenishments of the chamber by new magma pulses are likely responsible for the generation of a thin basal layer that crystallized the MML. To crystallize monomineralic magnetitite, the melt in this layer is supposed to be saturated in magnetite-alone, possibly due to the drop in pressure associated with melt ascent from a deep-seated staging chamber[57].

Upon emplacement into the chamber, the basal melt layer will cool until magnetite starts to crystallize. To gain further insights into this process, previous studies have attempted to model the distribution of Cr in magnetitite in one dimension. One study[1] considers growth of magnetite in a stagnant layer of melt. In this

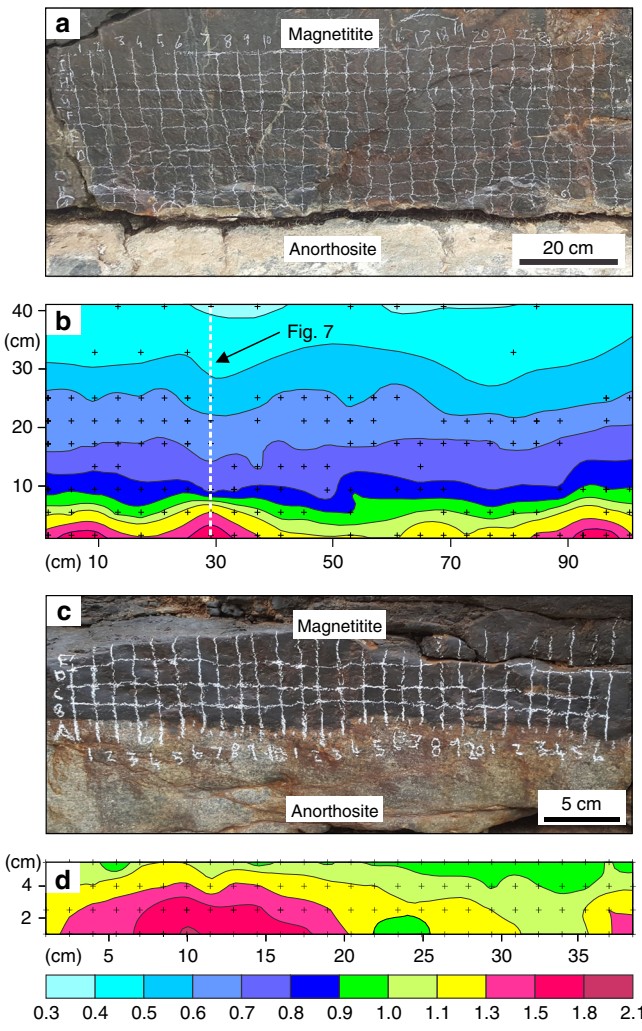

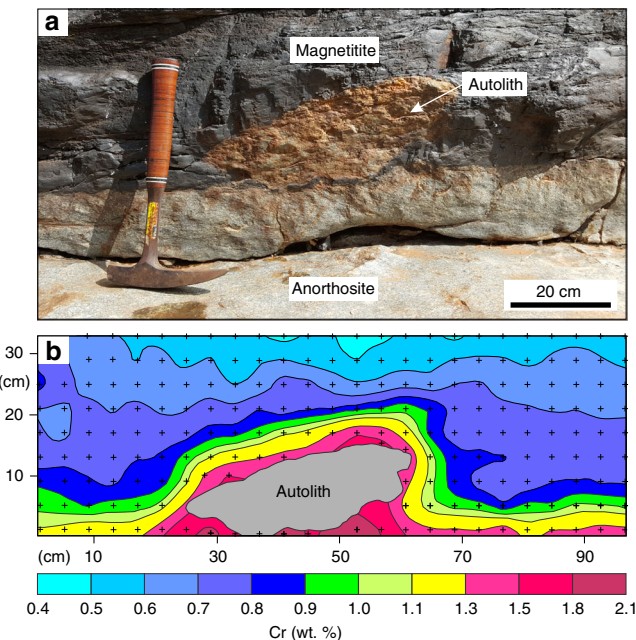

**Fig. 3 Two-dimensional geochemical maps of basal part of the Main Magnetite Layer with a planar basal contact.** Field photographs of studied areas in **a** and **c** are supplemented with their corresponding geochemical contour maps in (**b**) and (**d**), respectively. Black crosses indicate the positions of individual data points. Some data points were omitted because the analysis was not of sufficient quality for proper quantification. The maps reveal basal nodes showing a steep upwards and outwards depletion in Cr over short distances. These nodes represent places where magnetite starts to nucleate and grow providing evidence for the in situ crystallization of the MML. The vertical dotted line in (**b**), indicates the section used for modelling in Fig. 7. The outcrop is located at Magnet Heights, Eastern Bushveld Complex.

**Fig. 4 Two-dimensional geochemical map of basal part of the Main Magnetite Layer that contains a sub-rounded anorthosite autolith.** Field photograph of the section and the corresponding geochemical contour map are shown in (**a**) and (**b**), respectively. Black crosses indicate the positions of individual data points. Notice how the geochemical contours drape over the autolith, indicating in situ growth all along its outer surface. Several growth nodes appear directly underneath the autolith as well. The outcrop is located at Magnet Heights, Eastern Bushveld Complex.

model, growth of magnetitite is controlled by diffusion, leading to a profile of melt depleted in Cr directly at the crystal-liquid interface. In addition, occasional convective currents are involved to assist in supplying Cr to the growing magnetitite to prevent Cr from being depleted too rapidly[1]. However, later theoretical[58] and experimental[59] studies have shown that such a stagnant layer will experience vigorous convection from the very onset of crystallization, and the above process may therefore be physically unrealistic. Another study[35] envisages the crystallization of magnetite in a magma chamber stratified by double-diffusive convection. In this case, magnetite crystallizes in situ on the chamber floor, rapidly depleting the lowermost layer in terms of its Cr content. This causes Cr to diffuse from the overlying layers into the crystallizing basal layer, generally slowing the rate of Cr

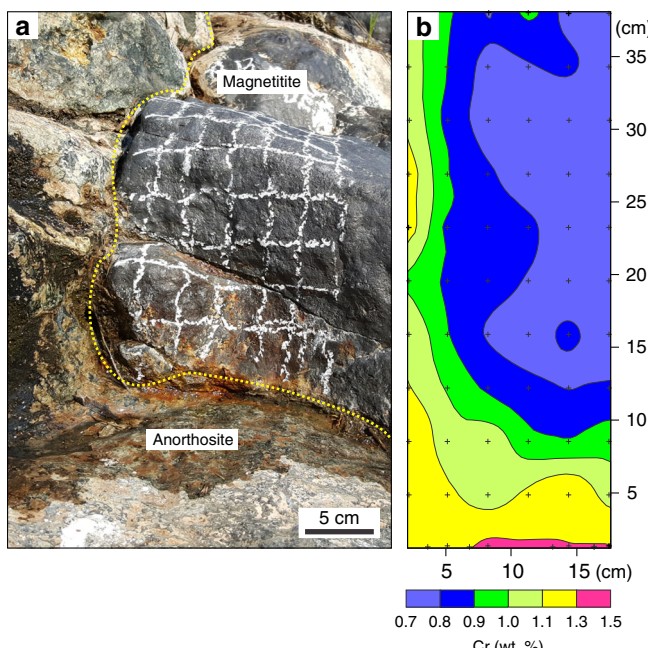

**Fig. 5 Two-dimensional geochemical map of basal part of the Main Magnetite Layer with a sub-vertical footwall contact.** Field photograph of the section and the corresponding geochemical contour map are shown in (**a**) and (**b**), respectively. Black crosses indicate the positions of individual data points. Geochemical contours remain subparallel with the footwall contact, indicating in situ nucleation and growth directly on the subvertical contact. The outcrop is located at Magnet Heights, Eastern Bushveld Complex.

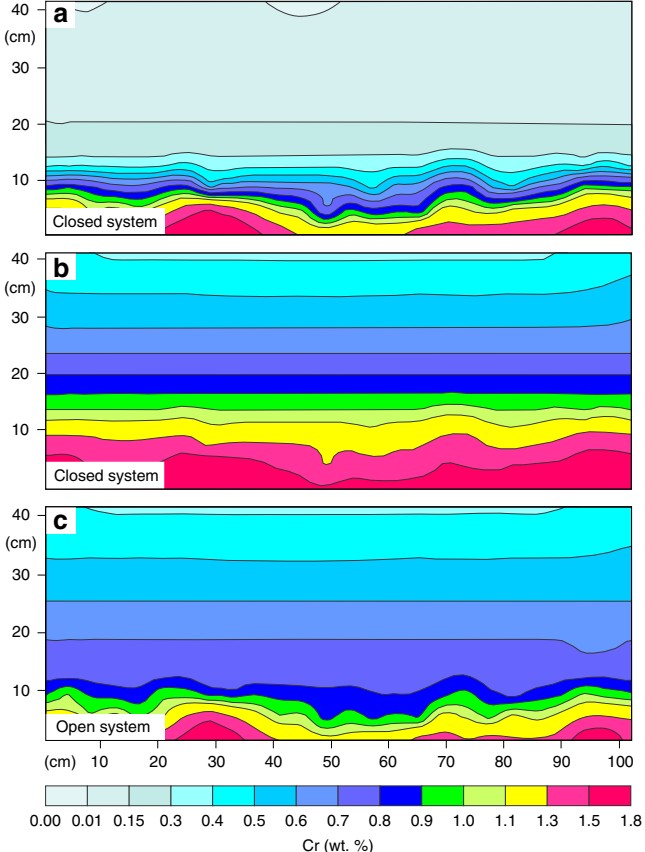

**Fig. 6 Two-dimensional geochemical modelling of Cr in the Main Magnetite Layer.** The two-dimensional modelling is based on the results shown in Fig. 3b and clearly illustrates the need for an open system to accurately describe the change in the Cr concentration ($D = 525$) in the MML. In (**a**), a closed system is assumed and a relatively thin basal layer is chosen (40 m) to accurately reproduce the Cr concentration around the growth nodes. However, higher up in the profile Cr is depleted much too rapidly compared with Fig. 3b. In (**b**), a thicker basal layer (130 m) more accurately reproduces the Cr concentration towards the top of the profile but the Cr concentration around the growth nodes is far too high. Only in an open system **c**, in which the basal layer grows incrementally from initially 20 m to beyond 57 m thick due to frequent magma addition, can the Cr-concentration accurately be reproduced in two-dimensions ("Methods"). Because undulations observed in the solidification front in the chemical map (Fig. 3b) are not statistically significant after about 20 cm of crystallization, a planar solidification front is assumed above this point in our modelling.

depletion. However, the presence of growth nodes shows that the MML did not simply grow in one direction (upwards), rendering a one-dimensional approach inaccurate. Our geochemical maps (Fig. 3–5) provide an opportunity to improve on these previous attempts[1,35] by performing modelling in two dimensions. We base our modelling on the two-dimensional distribution of Cr on the geochemical map of Fig. 3b. Our results reveal that the initial depletion around the growth nodes is much too fast, with the rate of depletion slowing down too strongly to be explained by the above models[1,35] or by the Rayleigh law alone. Although it is possible to produce the extremely rapid initial depletion by assuming a very thin basal melt layer (40 m) and simple Rayleigh fractionation, the depletion becomes much too rapid further upwards (Fig. 6a). By contrast, while it is possible to accurately reproduce the Cr concentration in the upper part of the profile by assuming a thick basal melt layer (130 m), this results in the Cr

concentration that is too high around the nodes (Fig. 6b). This issue can be resolved, however, if the basal melt layer is treated as an open system that develops incrementally via multiple magma additions that effectively mix with a melt in a basal layer. The incoming melt supplies Cr to the crystallizing layer, and, as it grows in thickness, the depletion rate of Cr gradually slows down. In contrast to what has been done before[35], this model accurately reproduces the two-dimensional distribution of Cr in magnetitite (Fig. 6c).

Yet another important outcome of the above modelling is that a combination of a steep chemical gradient of Cr depletion with much more constant V contents[60] is only reproducible if all rejected liquid from in situ growth nodes is returned into the overlying melt (Fig. 7; Methods). If this is not the case, then the rate at which the magma differentiates slows down dramatically[61]. The effect is much more pronounced on trace elements with very large partition coefficients (Fig. 8). If only as little as 3% of melt fails to return from in situ growth nodes to the overlying melt, a much thinner basal melt layer would be required to reproduce the Cr concentration in the MML compared with the case where fractionation is perfect (Fig. 7). Imperfect fractionation also causes trace elements with different compatibilities to behave more similarly. In such a thin boundary layer, V would also be rapidly depleted (Fig. 7a), while data from the MML shows no consistent increase or decrease in terms of the V concentration[60]. Perfect fractionation with a much thicker basal melt layer more accurately reproduces both the Cr and V concentrations (Supplementary Data 2). Since all liquid is returned from the solidification front to the liquid interior, no crystal mush would be able to develop at the chamber floor.

A good match with real chemical data can be produced when magnetite starts crystallising from a 6 to 27 m thick basal melt layer (assuming no interaction with the overlying resident melt) and with a $D$ value of 100 to 600 for Cr and 10 to 25 for V (Supplementary Fig. 2). Our final modelling (Figs. 6c, 7b) assumes a 20 m thick basal layer using a $D$ value of 525 for Cr and 20 for V. To reproduce the data, this basal melt layer is taken to grow in thickness incrementally to more than 59 m thick via the replenishment by Cr and V undepleted melt (Supplementary Fig. 2; Supplementary Data 2). In the modelling, we assume that a new pulse of melt enters the chamber after every centimetre of crystallization. Because the deposition of a centimetre-thick layer of crystals from a basal layer would barely affect its major element composition, it remains compositionally similar to the new pulses of melt entering the chamber. This will ensure effective mixing between the melt in the basal layer and the incoming melt from the very onset of magmatic recharge. The additional heat input from this process will temporarily halt the crystallization of magnetite, giving time for the compositional effects of magmatic recharge to spread evenly throughout the entire chamber before crystallization resumes.

## Discussion

There appears to be only one, hitherto largely neglected petrological process that can be reconciled with our geochemical data (Figs. 3–5) and modelling (Figs. 6–8). This is compositional convection governed by a gravitational instability of a thin liquid boundary layer around in situ growing crystals[4,58,59] (Fig. 9). During magnetite crystallization, Fe diffuses towards the crystal resulting in a thin boundary layer of a light rejected liquid. As crystallization proceeds, the boundary layer gradually increases in thickness and decreases in density as it loses Fe to the crystallizing magnetite growth nodes. Because our results suggest that magmatic differentiation starts operating after a magnetite node grows to only a couple of centimetres thick, the boundary layer

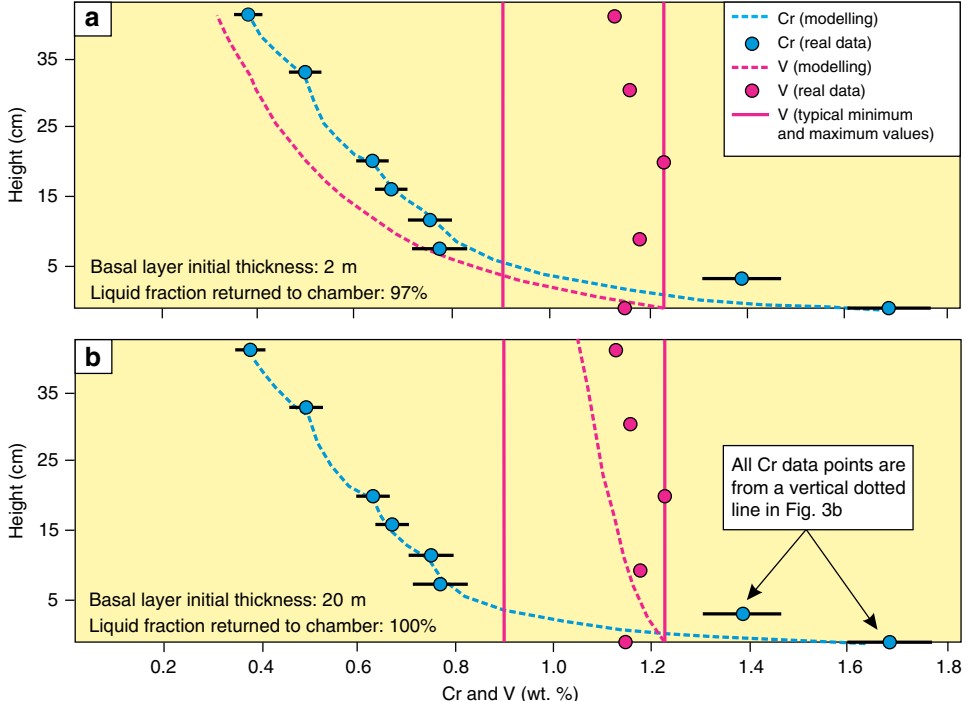

**Fig. 7 One dimensional geochemical modelling of Cr and V in the Main Magnetite Layer. a** One dimensional modelling of Cr and V in magnetite along a vertical dotted line in Fig. 3b. It is assumed that 97% of rejected melt is removed from the solidification front and returned to the main magma body. The basal melt layer grows incrementally due to frequent melt additions (Methods). Because fractionation is not perfect, the rate of Cr depletion is much slower, and a relatively thin (2 m) basal melt layer is needed to reproduce the depletion in Cr. In this case the behaviour of Cr and V are similar to each other, with both trace elements being rapidly depleted. However, while the Cr modelling curve matches reality, the modelled V concentration deviates dramatically from typical minimum and maximum values reported in the MML[60]. This approach is clearly not compatible with what is observed in reality. **b** In this case, it is assumed that perfect fractionation occurs so that 100% of rejected melt is immediately returned to the main magma body from the solidification front. It is now possible to reproduce the Cr concentration while the V concentration is much closer to reality using a 20 m thick basal melt layer. During the crystallization of massive magnetitite, near-perfect crystal-liquid fractionation must occur (Methods). V concentrations data plotted on this diagram are from Maila[60] as it was not possible to properly quantify V concentrations from our pXRF data. Horizontal black lines indicate the 2σ analytical uncertainty and are calculated by the instrument during analysis. However, because the Cr concentration is calculated from a Cr/V ratio (Methods), the Cr and V analytical uncertainties are compounded to obtain the final 2σ analytical uncertainty shown on the figure. Analytical errors for V are too small to show but standard deviation following the analysis of several samples are ~0.01 wt.%[60].

must reach sufficient buoyancy to convect away close to the very beginning of crystallization. To validate this inference, we performed calculations based on equations derived by Martin et al.[58] that allows the examination of the properties of a compositional boundary layer. These calculations suggest the boundary layer around a crystallizing magnetite exceeds the critical Rayleigh number for the onset of vigorous convection when it reaches a thickness of only 1.74 mm, with a compositional change across the boundary layer of about 3.02 wt.%. This corresponds to a density change across the boundary layer of 14.5 kg/m³ (Methods; Supplementary Data 3 and 4). At this stage, the boundary layer will obtain sufficient buoyancy to be released upwards into the main magma body, either as a constant stream of melt or as a series of plumes[4,58,59]. Their mixing with the overlying melt of the basal layer causes its chemical differentiation which is recorded in subsequently forming mm-thick shells of magnetite. When growth nodes eventually coalesce to form a planar solidification front, multiple plumes are released from random points on the front. Intermingling of the plumes results in vigorous convection and chemical differentiation of the basal layer[62] (Fig. 9). This rejection of unwanted chemical components causes such effective adcumulus growth of magnetite that the floor cumulates become totally solid directly at the crystal-liquid interface.

We thus conclude that three conventional processes of magmatic differentiation (Fig. 1) are not necessarily the dominant mechanisms of chemical differentiation in magmatic systems as deeply entrenched in most petrogenetic concepts[2,3,5,8]. A viable alternative is compositional convection caused by an unstable chemical boundary layer around in situ growing crystals at the chamber floor[4,58,59]. This process is unlikely to be unique to magnetitite layers. For example, field evidence for other cumulates of the Bushveld Complex, such as development of chromitite and orthopyroxenite layers along overturned portions of the chamber floor, attests to their in situ crystallization as well[39,40,63]. Similar to magnetitite, any other oxide-rich or mafic-ultramafic lithologies are expected to produce a buoyant boundary layer around the crystal liquid interface as has been demonstrated experimentally for in situ growing olivine[59]. In situ growth nodes may occur in these lithologies as well but cannot be detected due to the lack of sufficiently sensitive trace-element indicators of chemical differentiation in their minerals. We speculate therefore that this type of compositional convection may represent a very effective mechanism of magma evolution on Earth. Importantly, because of the very high buoyancy of a compositional boundary layer produced by crystallizing magnetite, a crystal mush is virtually absent in these oxide rocks, while it may only be a few metres thick in silicate rocks[40,43,64,65]. A lack of a thick mushy zone provides strong support for a classical 'magma chamber' paradigm (Fig. 1a) that has been underpinning models of the Earth's magmatism for over a century[1–8].

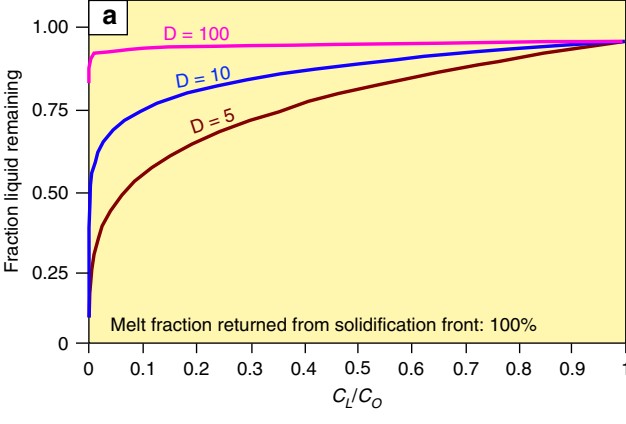

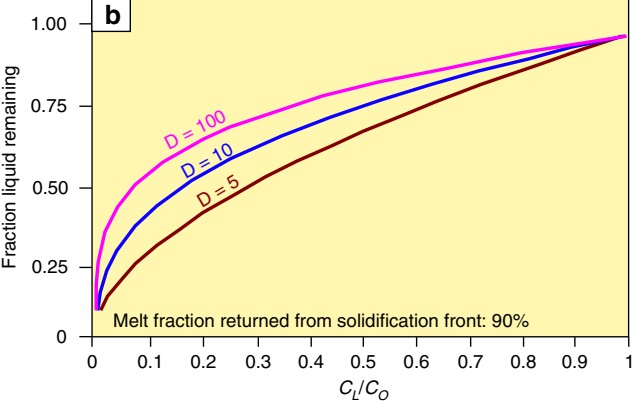

**Fig. 8 The comparative behaviour of compatible trace elements in magmatic systems with perfect and imperfect liquid fractionation.** During the crystallization of magma, compatible trace elements are rapidly depleted as the fraction liquid remaining decreases. **a** If perfect crystal-liquid occurs, the rate at which trace elements are depleted are most rapid, and magmatic differentiation is most effective. The varying rates of depletion is shown for three trace elements with varying values of $D$ (5, 10, and 100). **b** In this case, the fractionation of crystals and liquid is imperfect and 10% of the rejected liquid remains within the solidification front. This dramatically slows the rate of depletion of compatible elements with the effect being more dramatic for elements with higher compatibilities. This causes different trace elements with different compatibilities to behave more similar. Thus, to best describe the more constant V concentration compared with the extremely rapid depletion of Cr in magnetite from the Bushveld Complex (Fig. 7), the crystal-liquid fractionation needs to be near-perfect (Methods). $C_L$: Concentration of a particular trace element in the liquid. $C_O$: Initial concentration of a particular trace element in the liquid.

One important implication of our study is that it casts doubt on the universal validity of a recently advanced mushy reservoir paradigm[16,66] (Fig. 1b) that denies the existence of large and long-lived magma chambers in the Earth's crust and suggests that all magmatic systems, including layered intrusions, are predominantly composed of crystal-rich mushes. This concept is at odds with our data on solidification fronts in magnetite layers of the Bushveld Complex; the largest layered intrusion that is so widely employed for constraining petrological concepts. In addition, melt segregation and chemical differentiation in this novel paradigm is thought to be dominated by compaction[16,66]. However, the most recent petrographic studies indicate a notable lack of textural evidence for compaction in cumulate rocks of layered intrusions, including the Bushveld Complex[44]. This is perfectly reasonable since a very thin to nearly absent crystal mush in this[43] and other layered complexes[64,65] implies essentially nothing to compact. We hypothesize, therefore, that the

mere existence of the Bushveld Complex with its spectacular phase, modal and cryptic layering and giant mineral deposits[38], such as magnetitite layers up to hundreds of kilometres in length, is evidence that large, long-lived and largely molten magma chambers existed in the past, may operate right now and will likely emerge in the future life of our restless Earth.

## Methods

**Chemical analysis and quantification of data**. We have analysed an exposure of the MML located in the Magnet Heights area, Eastern Bushveld Complex (24° 50'14.83"S, 29°58'18.55"E) on grid patterns using a portable Niton XL3t XRF analyser. The instrument analyses an area with a diameter of 8 mm. A grid spacing was employed of 4 cm for the profiles in Figs. 3a, 4a, and 5a, and 1.5 cm for the profile in Fig. 3c. Each spot of the grid was screened with the portable XRF (pXRF) for ~60 s. The instrument was calibrated after every few hours of its use by means of its own built-in standards. To obtain quantitative data from the pXRF, we have performed the following recalculations. First, the Cr/V ratio of each analysis was determined. This is because the surface of the magnetite layers is not perfectly planar and it is not always possible to obtain proper contact with the pXRF for every analysis. As a result, the instrument underestimates the actual elemental concentration in nearly all cases. However, the analysis provides accurate measurements for elemental ratios as long as a fair amount of contact is maintained. This can be seen in terms of the constant V/Ti ratios recorded by our data (V/Ti ratios are also near-constant for in-house XRF analysis on pure magnetite separates from the MML, despite the fact that some deviation occurs from the ideal V/Ti ratio if grains of ilmenite are also included in the spot analysed). Spots with anomalous V/Ti ratios were omitted from the geochemical contour map. After calculation of the Cr/V ratios, the values are multiplied by 9.757 to obtain a quantitative Cr concentration in weight percent. This gives a fit with a linear function f(x)=x between the in-house and portable XRF data and an R-squared correlation coefficient of 0.97 (Supplementary Fig. 1). Geochemical contour maps were then constructed using Surfer Version 9.2.397. The average 2σ analytical uncertainty is 575 ppm, with the highest analytical uncertainty being 2350 ppm. To account for this uncertainty, geochemical contours on geochemical contour maps were spaced accordingly to ensure they are further apart than the maximum 2σ measured within each region. All pXRF data used in this study can be found in Supplementary Data 1.

**Equations for geochemical modelling**. We employ a modified version[67] of Langmuir's[61] equation for in situ crystallisation to reproduce the variation of both Cr and V in the MML:

$$C_L/C_O = (M_L/M_O)^{(f(D-1)/[D(1-f)+f])} \qquad (1)$$

where $C_L$ is the concentration of a trace element in the liquid, $C_O$ is the initial concentration of the trace element in the liquid, $M_L$ is the amount of liquid in the chamber, $M_O$ is the initial amount of liquid, f is the fraction of liquid that is ejected from the solidification front into the interior of the magma chamber, and $D$ refers to the partition coefficient for the trace element in question. This equation assumes that there is no trapped liquid in the solidification front. The equation also ignores the trapped liquid shift effect which is negligible considering the small amount of trapped liquid (zero to three percent) present in our modelling.

To perform modelling in two dimensions, the amount of magnetite contained within each contour on the geochemical contour map is measured. This measurement can then be used to calculate $M_L/M_O$ for Eq. (1) which is then used to determine the concentration at different coordinate positions on the contour map.

In an open system, changes in the Cr and V concentration due to melt addition is calculated using the following equation:

$$C_L = (M_i/M_t)(C_i) + (M_a/M_t)(C_a) \qquad (2)$$

where $C_L$ is the liquid composition after melt addition, $M_i$ is the amount of melt in the basal layer before melt addition, $M_t$ is the total amount of melt in the basal layer after melt addition, $C_i$ is the concentration of the liquid before melt addition, $M_a$ is the amount of melt added to the basal layer, and $C_a$ is the composition of the incoming melt. The latter value is calculated by dividing the highest Cr concentration measured[37] in the MML (4.8 wt. %) by the partition coefficient of Cr into magnetite. A spreadsheet used for the purpose of this modelling can be found in Supplementary Data 2.

In our final modelling (Figs. 6c, 7b), the initial thickness of the basal melt layer is 20 m (see the previous section). After every 100 cm$^2$ of magnetite crystallized in this part of the profile, a certain quantity of melt is added to the basal layer. For the first 1600 cm$^2$ of crystallization (or after the deposition of 16 cm of magnetite considering the section is 100 cm wide), 1.0 m of melt is added to the system, then 0.8 m of melt for the next 800 cm$^2$ of crystallization, followed by 0.54 m of melt for the next 800 cm$^2$ of crystallization, and finally 0.38 m of melt for the final 500 cm$^2$ centimetres of crystallization in the profile. A gradual decrease in the amount of melt added is necessary to explain the inflections observed in the Cr content (Fig. 7). At the top of the 40 cm high profile, the basal layer has grown to a

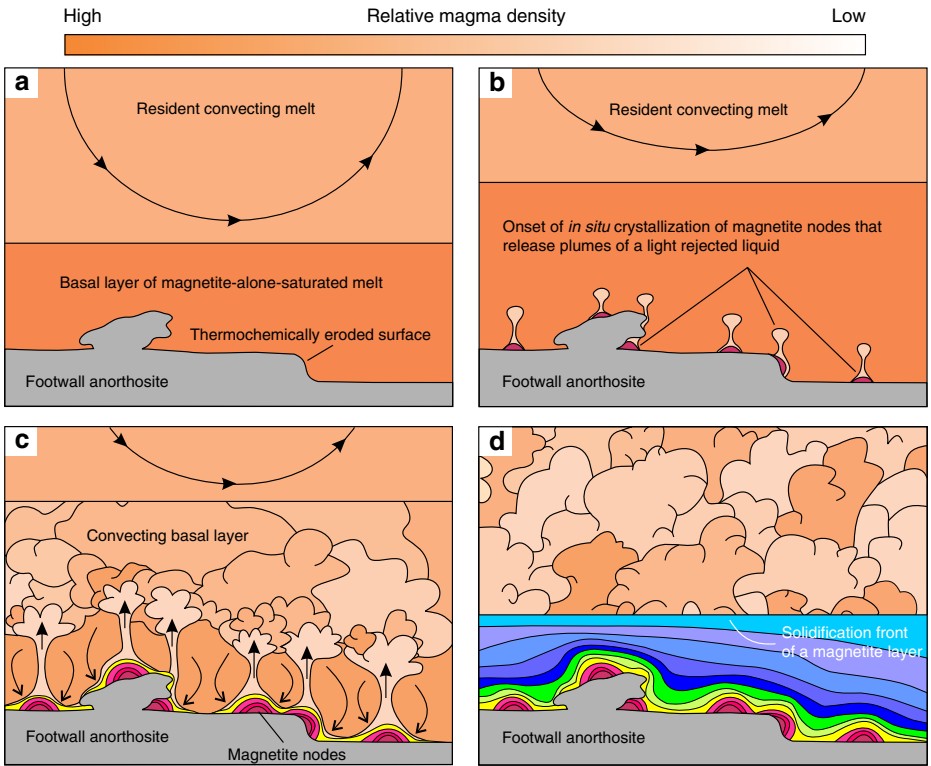

**Fig. 9 A physical model for chemical differentiation of the Main Magnetite Layer in the Bushveld Complex (not to scale). a** A magnetite-alone-saturated melt is emplaced as a basal layer along the chamber floor causing erosion of pre-existing footwall rocks. **b** On cooling, magnetite starts nucleating on the irregular surface of the anorthositic footwall. **c** Self-nucleation and crystal growth result in the formation of growth nodes. A residual, buoyant layer of liquid is produced around crystallizing nodes that migrates towards their top and is released into the overlying melt. **d** Nodes coalesce to form a planar solid floor; multiple plumes are released which leads to vigorous mixing in the basal layer. In each progressive step, the basal layer increases in thickness due to further addition of magnetite-alone-saturated melt into the chamber as indicated by geochemical modelling (Methods).

thickness of 48 m. After 1 m of magnetitite crystallization, it would have reached a height of 58 m.

**Choice of partition coefficients**. Because the V concentration of the melt parental to the Upper Zone of the Bushveld Complex is not known, we rely on experimental work to select a partition coefficient. The D values for V into magnetite ranges from 10 to 30 and is highly dependent on the oxygen fugacity of the melt[32]. Based on theoretical work, previous researchers have estimated the D of V for magnetite from the Bushveld to vary from 20 to 30 (ref. [30]). We use a D value of 20 because it reproduces the near-consistency of V as observed in vertical profiles through the MML.[60] more accurately compared with higher values. A D value of 20 implies that the melt had a V concentration of about 600 ppm.

Estimating the actual thickness of the basal layer is highly dependent on the partition coefficient used for Cr. The very rapid depletion of Cr at the base of the MML (up to more than 200 ppm/mm towards the top of growth nodes) requires an initially very thin basal layer of liquid. Based on experimental results, the D for Cr varies from 100 to 620 in basaltic melt[32]. Finding further constraints on the D value for Cr can be accomplished by studying the change in V in the layer. In vertical profiles across the MML in the Magnet Heights area, it was found that the V concentration may reach a maximum of 1.23 wt.%. The V concentration within the MML rarely drops below 0.90 wt.%[60]. Using a starting V concentration of 1.23 wt.% in magnetite, the V concentration should be kept above 0.90 wt.% during modelling. Using a low D value for Cr requires an initially thinner basal layer to describe the rapid rate of Cr depletion around growth nodes (Supplementary Fig. 2). In addition, a lower D value for Cr implies that the incoming melt has a higher concentration. This is because the incoming melt composition is calculated by dividing the maximum Cr recorded in the MML by the partition coefficient for Cr. A smaller amount of melt is thus required to model the retardation in Cr depletion upwards if Cr has a lower partition coefficient. Both the thinner basal layer and the smaller quantity of incoming melt leads to a more rapid depletion in V and a lower V concentration results at the top of the profile (Supplementary Fig. 2). Modelling is based on analysis selected from the dotted line in Fig. 3b but the profile has been extended to 100 cm high. At this height, the Cr concentration usually levels off at around 700 ppm[1], and the rate of melt addition is adjusted to mimic this effect in the modelling. Above 100 cm in the MML, an ~30 cm thick feldspar parting is present. No modelling is performed above the base of the parting. Under these conditions, only if a D value for Cr is used of more than 525

can V be kept above 0.9 wt.% assuming a D of 20 for V. By dividing the D for Cr in magnetite by the highest Cr content recorded in the MML, a Cr concentration of 91 ppm is obtained for the liquid prior to its formation. These parameters put a constraint on the minimum initial thickness of the basal layer of 20 m and are the parameters used in the final modelling (Figs. 6c, 7b).

**Properties of a compositional boundary layer**. During the in situ crystallization of a high density phase on the floor of a magma chamber, a thin boundary layer of liquid with a relatively low density develops around a growing crystal. This boundary layer progressively increases in thickness and decreases in density as the crystal grows until the boundary layer exceeds the critical Rayleigh number for convection. At this point the boundary layer breaks away from the growing crystal. Equations (3) and (4) below[58] allow the calculation of the compositional change across and the thickness of the boundary layer, respectively, at the very moment when the boundary layer becomes unstable and convection commences:

$$d_s = \left[ \frac{pvk_s^2}{C^3 \mathbf{g}\beta q} \left( L + C_p \frac{dT}{dS} \right) \right]^{1/4} \quad (3)$$

$$\Delta S = \left( \frac{v}{\mathbf{g}\beta k_s^2} \right)^{1/4} \left[ \frac{q}{pC \left( L + C_p \frac{dT}{dS} \right)} \right]^{3/4} \quad (4)$$

where ds is the compositional change across the boundary layer in weight fraction, $\Delta S$ is the boundary layer thickness, $p$ is the density of the melt, $v$ is the kinematic viscosity of the melt, $k_s$ is the diffusivity of the solute, $C$ is a dimensionless constant with a value of 0.1, $\mathbf{g}$ is acceleration due to gravity, $\beta$ is the compositional expansion coefficient, $q$ is the heat flux out of the chamber, $L$ is the latent heat of crystallization, $Cp$ refers to the specific heat capacity of the melt, and $dT/dS$ is the liquidus slope in temperature-solute space. The $p$, $v$, $Cp$, and $dT/dS$ were all calculated from using alphaMELTS software, version 1.4.1. An Fe-rich liquid composition that is considered to be parental for the Upper Zone[46] and which was slightly modified to have magnetite as the first liquidus phase was used in the MELTS run. The diffusivity of Fe in magma ($K_s$)[68] is taken as $10^{-11}$ m$^2$ s$^{-1}$ and the average $Cp$ of gabbro[69] (396,000 J kg$^{-1}$) is used. The compositional expansion coefficient ($\beta$) of magnetite is 0.36 (weight fraction$^{-1}$)[70]. A value for the heat loss through the basal layer ($q$) is taken so as to allow a crystallization rate of 1 cm per

year. The density change across the boundary layer was obtained from the MELTS run by choosing a composition that corresponds to the compositional change given by Eq. (4) above. The calculations performed to obtain the properties of the boundary layer can be found in Supplementary Data 3 as well as the MELTS run in Supplementary Data 4.

## Data availability

The authors declare that all relevant data are available within the article and its supplementary information files. Supplementary Data 1: Geochemical Data. Supplementary Data 2: Geochemical modelling. Supplementary Data 3: Compositional boundary layer properties. Supplementary Data 4: MELTS run.

## Code availability

All code used for the purpose of this study can be accessed in Excel spreadsheets in Supplementary Data 2–4.

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

## Acknowledgements
The original manuscript benefited from the constructive peer reviews by James Day, Brian O'Driscoll, Tim Ivanic, Sofya Chistyakova, Spike McCarthy, Olivier Namur, Ed Mathez, Tony Naldrett, Ed Ripley, Chris Hawkesworth, and Jean Bédard. We would like to thank Merrily Tau for his assistance in performing pXRF measurements in the field, Grant Cawthorn for providing us with magnetite samples and in-house XRF data for the construction of a calibration curve for the pXRF analysis, and Grant Bybee and Gelu Costin for their assistance with MELTS modelling. Sofya Chistyakova is thanked for helping us substantially improve the quality of our figures. The study was supported by research grants to R.L. from the National Research foundation (NRF) and Department of Science and Technology (DST)–NRF Centre of Excellence for Integrated Mineral and Energy Resource Analysis (CIMERA) of South Africa. Any opinion, finding and conclusion or recommendation expressed in this contribution is that of the authors and the DST-NRF CIMERA and NRF do not accept any liability in this regard. The clarity of the final version of this manuscript has been increased by comments of two official reviewers.

## Author contributions
W.K. performed pXRF analysis in the field, quantification of pXRF data, data processing, geochemical modelling, and wrote the first draft of the manuscript. R.L. and W.K. both contributed to the interpretation of data, formulating the original idea, discussing results and editing and writing the paper.

## Competing interests
The authors declare no competing interests.
