## [Peer Review File · Nature Communications]

REVIEWERS' COMMENTS:

Reviewer #1 (Remarks to the Author):

Recommendation:

The authors have done a good job of addressing the comments of the earlier reviewers. I think the paper is ready for publication (with some very minor changes), and I think it will be of interest to a wide audience of igneous petrologists. It examines a long running question about the exact nature of the crystal mush/magma interface in igneous intrusions in a new way. The paper presents an extensive dataset from a detailed study that can be used by other researchers to validate or refute their own models.

Comments:

1. The authors use a high distribution coefficient for Cr in magnetite, which seems to be one of the main criticisms of the earlier reviewers. Previously determined values for the distribution of Cr into magnetite cover a very wide range and I think the authors are justified in using the value they selected. Given the range of temperatures, pressures, and oxygen fugacities, at which Magnetite-Ulvospinel solid solutions crystallize, and the variation in %Mt - %Ulvo, it is not surprising that experimental and analytical estimates for the distribution coefficient will vary considerably.
2. Lines 29-31: The description of figures 1d and 1e in the text are the reverse of those in the figure and the figure caption.
3. Line 52: "common for" better as "common in"
4. Line 127: "as well all" should be "as well as"
5. Line 171: The crystallization of monomineralic magnetite layers may also be associated with a change in fO_2 .
6. Line 260: "affirm" better as "suggests"
7. Line 289: I assume the constant V/Ti ratio has been corrected for or examined with respect to the V-Ti peak overlap in X-ray data.
8. Line 351: "is taken that to allow" better as "is taken so as to allow"
9. Figure 2: Given that the difference in Cr and V is a major part of the paper, a plot of the V data, similar to the Cr data in figure 2 would be very useful. If not in the main paper, it could be included in the supplemental material.
10. Supplemental, page 1, 2nd para, line 6: "Magnet Heights are" this should be "Magnet Heights area"?
11. Supplemental, page 3, 2nd para, line 6: I understand that the model uses cm^2 because the layering is not horizontal, but the juxtaposition of meters of magma with cm^2 of magnetite makes it difficult to imagine at first. Just a sentence to say that the model is 100 cm wide so 1600 cm^2 is an average thickness of 16 cms would make it clearer to the reader.

H. R. Naslund

Reviewer #2 (Remarks to the Author):

I like this paper a lot. It is a fairly novel approach, analytically, to perform this sort of a study in situ, and has produced some extremely interesting and striking results. It has some major implications for what we understand about large magma storage in the crust - and goes back against the more recent surge towards crystal mushes, and proposes evidence for large volumes of melt instead. It is therefore, very topical and addresses a very fundamental aspect of igneous petrology. I therefore agree it is suitable for inclusion in Nature Communications.

I have read through the response to the previous reviews and the authors have done an excellent job of revising their manuscript accordingly. The discussion and use of particular D values appears to be well argued now and certainly acceptable for publication.

I appreciate the use of the supplementary movie to show the growth nodes in the model - this is particularly clear and I think represents a very plausible model consistent with the data. It does get around some of the unlikelyhoods in crystal settling models.

One question I have to the authors, and perhaps addressing this satisfactorily would help strengthen their arguments to any sceptics...

If these fronts grow as nodes, like in the video, and for the solidification front to become more planar with time, do we observe cases of much thinner magnetite layers with more uneven tops? If even very thin ones have planar tops, then does this make the model less likely? Or is the process scaleable?

Response to reviewers by authors (in red)

Reviewer #1 (Remarks to the Author):

Recommendation:

The authors have done a good job of addressing the comments of the earlier reviewers. I think the paper is ready for publication (with some very minor changes), and I think it will be of interest to a wide audience of igneous petrologists. It examines a long running question about the exact nature of the crystal mush/magma interface in igneous intrusions in a new way. The paper presents an extensive dataset from a detailed study that can be used by other researchers to validate or refute their own models.

Comments:

1. The authors use a high distribution coefficient for Cr in magnetite, which seems to be one of the main criticisms of the earlier reviewers. Previously determined values for the distribution of Cr into magnetite cover a very wide range and I think the authors are justified in using the value they selected. Given the range of temperatures, pressures, and oxygen fugacities, at which Magnetite-Ulvospinel solid solutions crystallize, and the variation in %Mt - %Ulvo, it is not surprising that experimental and analytical estimates for the distribution coefficient will vary considerably.

Thanks! Glad you agree with us on this point.

2. Lines 29-31: The description of figures 1d and 1e in the text are the reverse of those in the figure and the figure caption.

Thanks for pointing this out. It has now been fixed in the latest draft.

3. Line 52: "common for" better as "common in"

"Common in" has been replaced by "common for" as you suggested.

4. Line 127: "as well all" should be "as well as"

Thanks, this has been taken care of.

5. Line 171: The crystallization of monomineralic magnetite layers may also be associated with a change in fO_2 .

We agree that changes in fO_2 may cause the crystallization of magnetite. However, as we indicate in our paper, there are reversals associated with the formation of magnetite layers, indicating that magmatic recharge is probably responsible for their formation. Therefore we prefer a model that involves pressure reduction with magmatic ascent as we have indicated in our paper.

6. Line 260: "affirm" better as "suggests"

"Suggests" has been replaced by "affirm" in the new version of our paper.

7. Line 289: I assume the constant V/Ti ratio has been corrected for or examined with respect to the V-Ti peak overlap in X-ray data.

Yes, the portable XRF comes with its own built-in standards that allows the analysis of V and Ti simultaneously.

8. Line 351: "is taken that to allow" better as "is taken so as to allow"

Thanks, this has been fixed in the new version of our paper.

9. Figure 2: Given that the difference in Cr and V is a major part of the paper, a plot of the V data, similar to the Cr data in figure 2 would be very useful. If not in the main paper, it could be included in the supplemental material.

Unfortunately, we were unable to properly quantify V from our pXRF data. We have instead decided to plot V contents from a previous study from the same locality to show how the variation in V might typically look like.

10. Supplemental, page 1, 2nd para, line 6: "Magnet Heights are" this should be "Magnet Heights area"?

Thank! This error has been corrected.

11. Supplemental, page 3, 2nd para, line 6: I understand that the model uses cm² because the layering is not horizontal, but the juxtaposition of meters of magma with cm² of magnetite makes it difficult to imagine at first. Just a sentence to say that the model is 100 cm wide so 1600 cm² is an average thickness of 16 cms would make it clearer to the reader.

We have added a sentence in the new version of our paper to hopefully make this clearer as you suggested.

H. R. Naslund

Reviewer #2 (Remarks to the Author):

I like this paper a lot. It is a fairly novel approach, analytically, to perform this sort of a study in situ, and has produced some extremely interesting and striking results. It has some major implications for what we understand about large magma storage in the crust - and goes back against the more recent surge towards crystal mushes, and proposes evidence for large volumes of melt instead. It is therefore, very topical and addresses a very fundamental aspect of igneous petrology. I therefore agree it is suitable for inclusion in Nature Communications.

I have read through the response to the previous reviews and the authors have done an excellent job of revising their manuscript accordingly. The discussion and use of particular D values appears to be well argued now and certainly acceptable for publication.

I appreciate the use of the supplementary movie to show the growth nodes in the model - this is particularly clear and I think represents a very plausible model consistent with the data. It does get around some of the unlikelihoods in crystal settling models.

One question I have to the authors, and perhaps addressing this satisfactorily would help strengthen their arguments to any sceptics...

If these fronts grow as nodes, like in the video, and for the solidification front to become more planar with time, do we observe cases of much thinner magnetite layers with more uneven tops? If even very thin ones have planar tops, then does this make the model less likely? Or is the process scaleable?

Thanks for your kind words regarding our paper! The thinnest magnetite layers are about 10 cm thick. It is interesting to note that they do indeed also possess planar tops! However, our results seem to indicate that the solidification front may already become planar after just 10 cm of crystallization, explaining why you do not see undulating tops even within these layers.